# Earliest signs of life on land preserved in ca. 3.5 Ga hot spring deposits

Tara Djokic[1,2], Martin J. Van Kranendonk[1,2,3], Kathleen A. Campbell[4], Malcolm R. Walter[1] & Colin R. Ward[5]

The ca. 3.48 Ga Dresser Formation, Pilbara Craton, Western Australia, is well known for hosting some of Earth's earliest convincing evidence of life (stromatolites, fractionated sulfur/carbon isotopes, microfossils) within a dynamic, low-eruptive volcanic caldera affected by voluminous hydrothermal fluid circulation. However, missing from the caldera model were surface manifestations of the volcanic-hydrothermal system (hot springs, geysers) and their unequivocal link with life. Here we present new discoveries of hot spring deposits including geyserite, sinter terracettes and mineralized remnants of hot spring pools/vents, all of which preserve a suite of microbial biosignatures indicative of the earliest life on land. These include stromatolites, newly observed microbial palisade fabric and gas bubbles preserved in inferred mineralized, exopolymeric substance. These findings extend the known geological record of inhabited terrestrial hot springs on Earth by ~3 billion years and offer an analogue in the search for potential fossil life in ancient Martian hot springs.

[1] Australian Centre for Astrobiology, PANGEA Research Centre and School of Biological, Earth and Environmental Sciences, University of New South Wales, Kensington, New South Wales 2052, Australia. [2] Australian Research Council Centre of Excellence for Core to Crust Fluid Systems (CCFS), Macquarie University, New South Wales 2109, Australia. [3] Big Questions Institute, University of New South Wales Australia, Kensington, New South Wales, 2052 Australia. [4] School of Environment, University of Auckland, Private Bag 92019, Auckland 1142, New Zealand. [5] School of Biological, Earth and Environmental Sciences, University of New South Wales Australia, Kensington, New South Wales 2052, Australia. Correspondence and requests for materials should be addressed to T.D. (email: t.djokic@unsw.edu.au).

Exceptional preservation of biosignatures in Archaean rocks provides unique insight into the early history of life on Earth and offers a guide in the search for ancient biosignatures on Mars. Some of Earth's earliest convincing evidence of life is from the ca. 3.48 Ga Dresser Formation of the North Pole Dome, Pilbara Craton, Western Australia, which previously had been interpreted as a marine environment[1,2]. However, recent evidence indicates that the Dresser Formation was deposited within a volcanic caldera affected by voluminous hydrothermal fluid circulation[3–7]. These strata comprise two horizons of silicified sedimentary rocks alternating with pillowed to massive metabasalts[2–5]. The formation is exceptionally well preserved for its age, exhibiting low-strain and low-grade metamorphism, specifically prehnite–pumpellyite to lower greenschist facies[4,6]. The lowest sedimentary horizon of the Dresser Formation, here referred to as DFc1, is a fossiliferous unit (∼4–60 m thick), well exposed for ∼14 km along the eastern flank of the North Pole Dome[2,6]. DFc1 consists of grey and white layered chert, with subordinate volcaniclastic sandstone, jasplitic chert, bedded carbonate and stromatolites[6]. Underlying hydrothermally altered komatiitic metabasalts are transected by a dense network of silica (microquartz) ± barite ± pyrite ± organic matter-bearing hydrothermal veins that were contemporaneous with sediment accumulation, as they disperse into, but do not pass through, DFc1 (refs 3–9).

Within DFc1 and its hydrothermal chert-barite veins is a suite of known biosignatures that include stromatolites[1,2,5,6,10,11], putative microfossils[11] and fractionated carbon[8,9,11] and sulfur[12] isotopic evidence. Tentative links between life and circulating hydrothermal fluids have been inferred from stromatolites proximal to, and interbedded with, hydrothermal barite vein deposits[5,10]; carbon isotopic signatures from organic matter in black silica veins and bedded cherts[8,9,11]; sulfur isotopic measurements from microscopic pyrite in hydrothermal barite veins[12]; and fluid inclusion data from shallow subsurface hydrothermal quartz veins that was used to indirectly suggest the presence of surface hot springs[7]. Missing from the caldera model, until now, was direct evidence for hot spring fluids debouching onto the land surface to form distinctive siliceous sinter deposits, and their unequivocal link with life.

Modern, terrestrial, siliceous hot spring deposits (sinters) display a rich diversity of sedimentary facies derived from combined biogenic and abiogenic activity[13]. Typically, silica-rich thermal fluid of near-neutral pH, alkali-chloride composition discharges from hot springs and precipitates opaline silica on available biotic and abiotic surfaces to build up a broad sinter apron[14]. Sinters are diagnostic of terrestrial geothermal fields, displaying various textures indicative of proximal vent to distal discharge-apron facies[15–17] corresponding to evaporative cooling of thermal water in channels, pools and terraces (100 °C to ambient). Distinctive temperature-dependent, biotic communities flourish today, including microbial mats and biofilms[15–17]. Geyserite is a type of sinter formed exclusively in and around the peripheries of vent-related (75–100 °C) geyser and spouter mounds as well as along the rims of hot spring-pool source areas, formed by the action of splashing and surging of hot silica-rich water[13,18,19]. Geyserite was initially thought to be abiogenic[13]; however, microstructural development of geyserite may be influenced by microorganism biofilms acting as a substrate for silica precipitation[14]. Commonly, evidence of biofilms can be lost in hot vent areas due to rapid silica infill and replacement[19], whereas thicker microbial mat textures may be well preserved at lower temperatures away from the vent[20]. Currently, geyserite has been recorded only from rocks as old as the Devonian, that is, ca. 400 Ma[19].

Here we provide stratigraphic, petrographic and geochemical evidence from newly discovered, finely laminated siliceous rocks in the Dresser Formation that we interpret as hot spring-related sinter, including geyserite. Masses of barite with isopachous layering that occur at the tops of large hydrothermal veins, which directly underlie these surface hot spring deposits, were previously interpreted as seafloor mounds, but here are re-interpreted as the mineralized remnants of the hot spring pools or vents. Importantly, several new potential biosignatures, including stromatolites, were observed within these deposits. These findings extend the record of inhabited hot spring deposits by ∼3 Ga, indicate that life colonized land ∼580 million years earlier than previously thought[21], and have implications for the search for life on Mars.

## Results

**Dresser Formation geyserites.** Distinctive microlaminated siliceous rocks observed at three DFc1 localities, ∼2 km apart, are interpreted here as geyserite (Supplementary Fig. 1). These deposits contrast markedly with all other finely laminated sedimentary rocks in DFc1 (refs 2,5) as they contain an order of magnitude finer scale, dense lamination and distinct mineralogical and petrographic features. DFc1 inferred geyserite deposits are 2 mm–3 cm thick, laterally restricted horizons of varied textures—planar to wispy (locality 1S: Supplementary Fig. 1), or stratiform to columnar–botryoidal (locality 16N: Supplementary Fig. 1; Fig. 1a–d)—composed of very fine-grained (1–10 μm), siliceous, alternating light/dark microlaminae, 2–30 μm thick (Fig. 1e). Contacts between the light/dark laminae are well-defined, but gradational on a micron-scale. In the best-preserved sample, from locality 16N, columns and botryoids are overlain by stratiform laminae (Fig. 1b,c). The edges of the columns and botryoids consist of overhanging laminae that pass diffusively into troughs filled with slightly coarser-grained (10–40 μm), equigranular (unlaminated) microquartz that resemble geyserite cornices[13]. Microlamination may be continuous for up to 5 mm across a number of columns and botryoids, or discontinuous, with local cross-laminae displaying onlap/offlap relationships relative to underlying laminae (Fig. 1b). Small-scale, syn-depositional slumps are locally preserved in the very fine laminae (Fig. 1c). In one example, a well-developed set of stratiform layering is overgrown by botryoidal–columnar laminae that wrap around and extend downward, underneath the eroded edge of the stratiform layer, displaying botryoids that protrude horizontally and then downward around the lamina set (Fig. 1d).

The macro- and micro-scale textures displayed by the inferred Dresser geyserite are directly analogous to features observed in modern geyserite[13,18,19] (Fig. 1f–h) and contrast with hydrothermal vein textures, which typically display colloform banding composed of macroquartz crystals growing inward from the cavity rim and which lack erosion/re-deposition features or troughs separating botryoids[7,22]. Microlaminae within the inferred geyserite are cut by barite crystals, indicating deposition during hydrothermal activity within DFc1, as barite is absent from the overlying lithology[6]. The inferred Dresser geyserite is discounted as a post-depositional feature due to the presence of angular, millimetre-sized geyserite rip-up clasts (sintraclasts)[23] in a unit of edgewise conglomerate, supporting a synsedimentary interpretation (locality 23S: Supplementary Figs 1 and 5). Sintraclasts form by reworking and re-deposition of geyserite or apron sinter in fluvial settings, commonly during alternating wet and dry intervals on outflow aprons, the extent of which can span up to many tens of metres away from a vent[23]. The Dresser sintraclasts are interbedded with inferred fluvial deposits that include pebble to cobble conglomerate with rounded chert

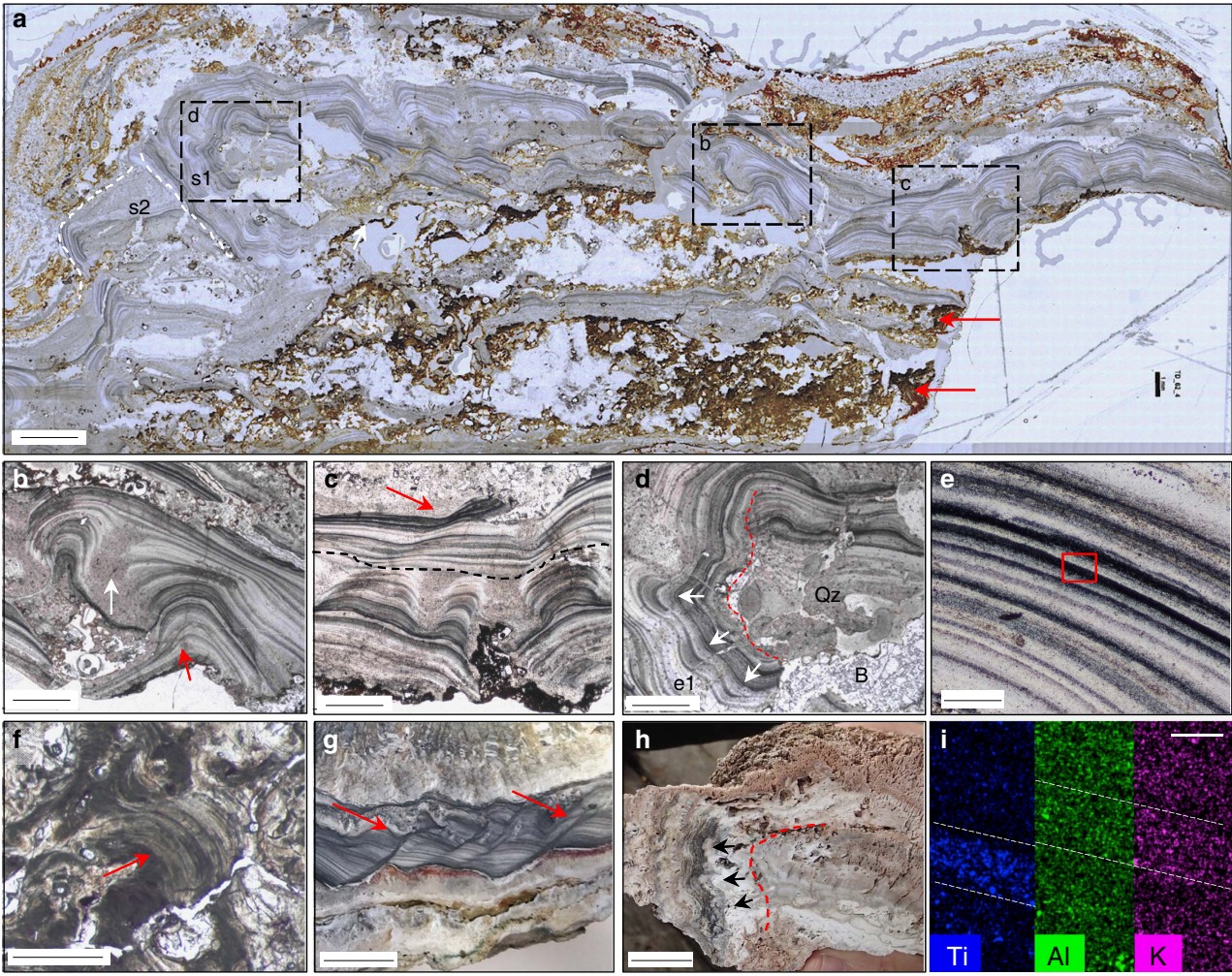

**Figure 1 | Comparison of Dresser geyserite with modern examples.** Scale bar measurements indicated. (**a**) High resolution gigapan image of Dresser geyserite. Inset boxes are figure parts (**b,c,d**). Laminae overgrowth stages; s1 and s2 represented by white dashed lines. Ferruginous material (red arrows) contains inferred gas bubbles; see Fig. 5. Scale bar, 2 mm. Micrographs in PPL (**b**–**f**). (**b**) Botryoidal textures display laminae onlap/offlap (red arrow), separated by siliceous equigranular troughs (white arrow) overlain by fine, planar laminae (scale bar, 1 mm). (**c**) Botryoidal–columnar textures overlain by planar (black dashes), slumped (red arrow) laminae. Scale bar, 1 mm. (**d**) Overgrowth (e1) with outward and downward facing botryoids (white arrows). Quartz (Qz) and barite (B), infill and cross-cut laminae (scale bar, 1 mm). (**e**) Close-up of light/dark microlaminae in Dresser geyserite. Inset box of figure part (i). Scale bar, 50 μm. (**f**) Modern geyserite with botryoidal microlaminae (red arrow), Geysir, Iceland. Analogous to (**b**). Scale bar, 1 mm. (**g**) Slumped laminae of <100-year-old geyserite, Geyser Valley, New Zealand. Analogous to **c**. Scale bar, 1 cm. (**h**) Pool rim overgrowth of geyserite with outward facing botryoids (arrows), Geyser Valley, New Zealand. Analogous to **d**. Scale bar, 2 cm. (**i**) SEM-EDS element maps showing light bands enriched in K–Al alternating with dark bands enriched in Ti, identified as kaolinite + illite and anatase, respectively, from Raman spectroscopy and XRD analysis; see Supplementary Figs 2–6 (scale bar, 50 μm).

pebbles, and edgewise conglomerate containing long, but thin (aspect ratios of 40:1), locally slightly bent, siliceous clasts stacked in vertical arrays (Supplementary Fig. 2). These sedimentologic relations are typical of high-energy events in very shallow water and are consistent with formation as fluvial deposits in the outflow channels of hot spring pools[19].

Somewhat thicker laminae in the Dresser geyserite (2–30 μm), compared to modern geyserite (500 nm–4 μm thick)[13], may be attributed to diagenesis, whereby very fine laminae are destroyed in the transition from opal-A to microquartz[19,24], as seen in Late Jurassic geyserite examples[19]. Significantly, the Dresser geyserite deposits are found directly overlying mineralized barite veins emblematic of subsurface hydrothermal fluid pathways, described in detail below (Figs 3 and 4 and Supplementary Fig. 1).

To distinguish the inferred geyserite deposits from other silica-replaced lithologies, such as microstromatolites[13], or silicified sediments or travertines[19], the composition of the siliceous

light/dark laminae was examined in all Dresser samples using scanning electron microscopy-energy dispersive spectroscopy (SEM-EDS), Raman spectroscopy and X-ray diffraction (XRD) (mapping, point and bulk analyses; Supplementary Figs 3–7). SEM-EDS element maps show higher concentrations of Al and K within lighter bands, which contain singular and clustered lath-like flakes, 2–30 μm in size. Flakes are composed primarily of Si, O, Al and K with minor Na, Ba and Mg dispersed within a siliceous matrix (Si, O and very minor Al). XRD patterns indicate that the Al- and K-rich laths and flakey aggregates are composed of intermixed kaolinite ($Al_2Si_2O_5(OH)_4$) and illite ($(K,H_3O)$ $(Al,Mg,Fe)_2(Si,Al)_4O_{10}[(OH)_2,(H_2O)]$). SEM-EDS element maps display concentrations of Ti within darker laminae, revealed by SEM as very fine (2 μm) crystal grains composed dominantly of Ti and O, locally with trace amounts of K, Al and Na. However, these trace elements, as well as large silicon peaks, are considered to derive from the microquartz matrix (Si, O with minor K and

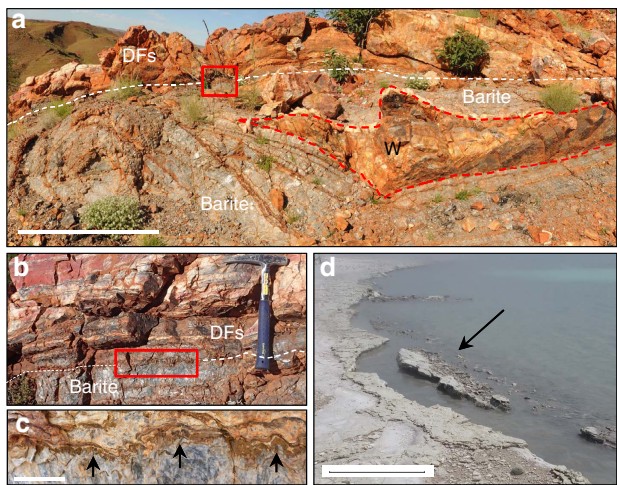

**Figure 2 | Sinter terracettes and microbial palisade fabric.** Scale bar measurements indicated. (**a**) Dresser terracettes (red arrows) with preserved primary porosity (green arrow) and a horizon containing Dresser stratiform geyserite (black arrow). Scale bar, 1 cm. Inset box of **c**. displays palisade fabric. (**b**) >1,800-year-old sinter terracettes (red arrows) with preserved primary porosity (green arrow) from a sinter buttress at Te Kopia, New Zealand. Scale bar, 1 cm. Micrographs in XPL of (**c**) Dresser palisade fabric oriented vertical to bedding (scale bar, 1 mm) and (**d**) close-up (scale bar, 250 μm). (**e**) Sinter with preserved palisade fabric, Te Kopia, New Zealand. Scale bar, 1 mm.

**Figure 3 | Large isopachous barite masses as mineralized hot spring pools.** Scale bar measurements indicated. (**a**) Strongly curving isopachous barite veins envelop a chert wedge (W), overlain by sedimentary units (DFs). Scale bar, 1 m. Inset box of (**b**) isopachous layered barite underlies DFs that include stratiform geyserite. Inset box of (**c**) barite crystal tops growing upward into DFs (arrows). Scale bar, 2 cm. (**d**) Modern collapsing hot spring-pool lip edge, shoreline of Lake Rotokawa, Rotokawa geothermal area, New Zealand. Scale bar, 0.5 m.

Al ± Na). At 20 kv, interaction/penetration of the beam is a minimum of 2–3 μm, whereas many of the Ti-grains are of a similar size (that is, ~2 μm), or larger. Thus, it is unlikely that a discrete signal was acquired solely from the Ti-rich grains. Raman spectra showed major peaks at 143 and 638, indicative of anatase ($TiO_2$), and a major peak at 464, indicative of quartz. Mineral confirmation XRD patterns indicate that dark laminae contain abundant, micron-scale, anatase crystals dispersed in a matrix of microquartz. Equigranular troughs between laminated botryoids contain relatively equal proportions of Ti, K and Al (Supplementary Fig. 7).

Both kaolinite + illite and anatase are documented alteration minerals in the upper level of geothermal fields, for example, at Sulphur Springs, New Mexico[25], and the Soufriere Hills volcano, Montserrat[26]. Anatase has also been reported in Jurassic geyserites[19]. Kaolinite + illite is a diagnostic alteration mineral suite in shallow, advanced to intermediate argillic alteration (~<120 °C) zones of high-sulfidation epithermal systems[24], including that of the Dresser Formation, which displays steam-heated acid-sulfate kaolinite + illite alteration of underlying pillow basalts[4]. Precipitation of anatase is favoured in near

neutral to alkaline pH[27], which is consistent with formation of geyserite in modern low temperature (~<100 °C), near neutral alkali-chloride thermal springs[19]. The anatase is unlikely to be a retrograde alteration product of rutile (high-temperature polymorph of $TiO_2$), as the latter is preserved in the higher temperature phyllic alteration zone within underlying basalts around the Dresser barite mine[4], located <3 km from sites containing inferred geyserite. Temperature constraints in DFc1 are indicated from preservation of anatase, which irreversibly transforms to rutile above 400 °C (ref. 28); presence of stable kaolinite + illite mineral assemblages, indicative of temperatures <120 °C; and fluid inclusion data from DFc1, which indicate relatively cool (~120 °C) water temperatures under low-confining pressures near the palaeosurface[7].

**Sinter terracettes.** At locality 1S (Supplementary Fig. 1), stratiform geyserite is overlain by a 3 cm thick unit composed of siliceous, millimetre-thin laminae that form ~1 cm diameter, low-amplitude (<2 cm), asymmetrical convex ridges (Fig. 2a). The laminae in these ridges are stacked into what superficially resemble climbing ripples, but differ in that thicker laminae appear on the down-current side. The convex laminae in cross-section resemble sinter terracettes (smaller-scale subsets within sinter terraces) from the mid- to distal-apron facies of hot springs, displaying the primary porosity and microtextures comparable with more recent, microbially derived examples[19] (Fig. 2a,b).

**Mineralized hot spring pools.** New observations are presented here with respect to the formation mechanism of large barite masses found at the upper tips of black silica + barite veins, which are relevant in interpreting a terrestrial hot spring setting for DFc1. Subspherical (5–20 m diameter) masses of coarsely crystalline, isopachous, hydrothermal barite (+ pyrite) occupy the uppermost parts of hydrothermal veins where they contact DFc1 sedimentary deposits. Previously, these barite masses were interpreted as baritized, diapiric gypsum bodies[2], and later were inferred to be primary 'barite mounds' formed on the seafloor[3]. However, new observations suggest that at least some of these barite masses represent the mineralized remnants of terrestrial hot spring pools and associated shallow subsurface hydrothermal plumbing.

Significantly, large hydrothermal barite masses immediately underlie two of the geyserite localities described here, out-cropping at the top of 10–20 m wide, ~1 km deep silica + barite hydrothermal veins that cut their way up into the base of the finely layered sedimentary succession (Supplementary Fig. 1). The barite masses typically consist of multiple, thick, distinctly curved

isopachous layers of coarsely crystalline barite (Fig. 3a) with crystals up to 10 cm long that consistently point upwards and outwards towards the edges of the masses. Typically, sets of barite crystals are separated by thin pyrite laminae from which sulfur isotopic values point to microbial disproportionation[12]. At locality 1S (Supplementary Fig. 1), a large hydrothermal barite mass displays distinct sets of isopachous barite layers with strongly curving geometries that envelop a wedge-shaped block of layered chert-barite derived from the overlying sedimentary succession (Fig. 3a). The uppermost barite layer has crystal tops that project into the base of the overlying sedimentary unit containing geyserite and siliceous sinter (Fig. 3b,c). Together, these observations suggest that barite mineralization developed beneath a collapsing, but semi-lithified sedimentary crust containing localized geyserite and siliceous sinter. This interpretation is supported by observations made 3 km north of the Dresser Mine, where a 10 m long × 1 m thick tilted panel of bedded chert + microbialites + vein barite, together with angular to rounded blocks of hydrothermal barite and chert, form a megabreccia (devoid of geyserite) that fills what was a large subspherical cavity with steeply dipping walls that cuts down through bedded chert (see for comparison Fig. 13, p. 215 of ref. 5). Formation of the megabreccia occurred during sediment accumulation, as demonstrated by bedded chert that overlies the cavity.

These barite masses occur at the uppermost tips of hydro-thermal feeder veins along faults (that is, hydrothermal fluid conduits), and some are found immediately beneath overlying strata with known geyserite and siliceous sinter deposits. Therefore, these large enveloped and tilted sedimentary blocks are interpreted as collapsed hot spring terraces or pool margins such as those observed in modern geothermal areas (Fig. 3d).

The geometry and isopachous nature of these barite masses may be compared to fossilized travertine deposits of Lake Bogoria, Kenya, where isopachous carbonate layers systematically line and fill subterranean cavities of the former hot spring pools (see Fig. 4, p. 806 in ref. 29). While barite is not present in the Lake Bogoria example, the textures are equivalent. Although rare, terrestrial hot spring barite is known to precipitate alongside silica[30], but no reported modern examples host the large quantities of barite found in DFc1. In summary, these data suggest that the isopachous barite masses represent the mineralized remnants of hot spring pools at the uppermost parts of the geothermal plumbing system (Fig. 4).

**Biosignatures in Dresser hot spring deposits.** Horizons containing intergrown hydrothermal microquartz and barite interspersed with Fe-oxyhydroxides (the latter a product of Tertiary weathering of primary pyrite) occur between sets of the fine light/dark siliceous microlaminae within the Dresser columnar–botryoidal geyserite at locality 16N (Fig. 1a and Supplementary Fig. 8). Contained within these horizons are numerous circular to sub-circular structures, ~200 µm in diameter, filled by microquartz and barite, but lined with fine-grained anatase and internal anatase crystal splays that fan inwards from the margins towards the centre of the structures (Fig. 5a–c). The infilling microquartz and barite cuts across the anatase-lined walls of the structures, indicative of their early formation during DFc1 hydrothermal activity (Fig. 5a,d).

These circular structures are discounted as fluid inclusions based on their larger size (20× larger than typical fluid inclusions at 10 µm diameter) and their occurrence in a polymineralic matrix[7]. Rather, the structures are interpreted as gas bubbles[31]. Many gas bubbles, both biogenic and abiogenic, are subject to deformation caused by their host lithology[31,32], yet most Dresser bubbles retain a circular outline with only a few showing collapsed margins. Their excellent preservation suggests growth within, and perhaps partial passage through, an elastic medium that also trapped them before they reached the surface and burst. The only known medium in hot spring settings with such properties is exopolymeric substance (EPS)[31] derived from microbial mats/biofilms that flourish on the mid-aprons of modern sinter terraces (Fig. 5e). Entrapment of bubbles in EPS would allow for retention of the bubble structure while allowing for internal, inward radiating crystallization of anatase during hydrothermal alteration of the biofilm.

Alternately, bubbles observed floating on the surface of hot spring pools from Mammoth Hot Springs at Yellowstone National Park were reported as having preserved their shape via calcite crystallization[33]. However, such crystals radiate outward from bubble surfaces and thus contrast with the internal, inward radiating anatase crystals in the Dresser bubbles. Rather, entrapment in EPS would retain the bubble structure while allowing internal, inward radiating crystallization.

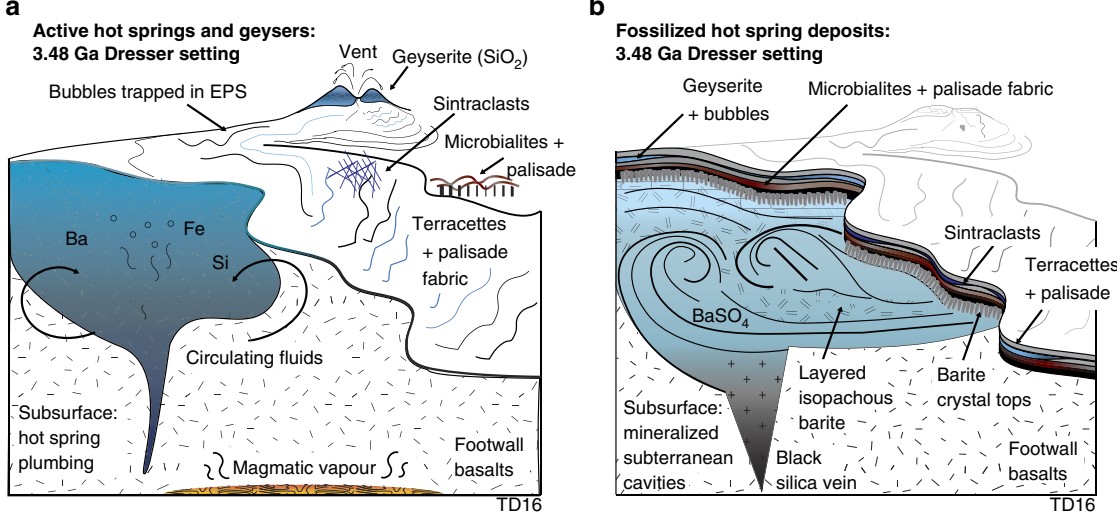

**Figure 4 | Schematic model of active Dresser hot spring system and its fossilized mineralized remnants.** (**a**) Proximal to distal hot spring facies, with spring vent fed by subsurface hydrothermal fluids. (**b**) Preserved sequence of hot spring facies deposits, geographically patchy in nature, with spring vent infilled by late-stage crystallization of barite.

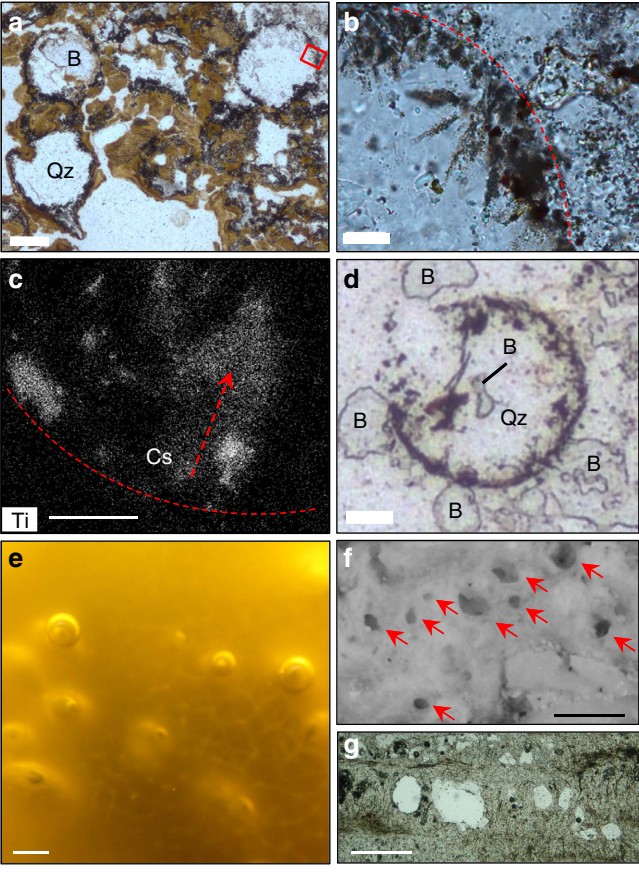

**Figure 5 | Inferred gas bubbles.** Scale bar measurements indicated. Micrographs in PPL (**a**,**b**). (**a**) Spherical to subspherical inferred bubbles filled with quartz (Qz) and barite (B). Scale bar, 90 μm. Inset box of (**b**) inward radiating anatase crystal splays from bubble rim (dashed line). Scale bar, 22 μm. (**c**) SEM-EDS element map data of Ti concentration in crystal splay (Cs) along bubble rim (dashed line). Scale bar, 9 μm. (**d**) Quartz (Qz) and barite (B) cutting bubble wall. Scale bar, 22 μm. (**e**) Oxygen bubbles in modern EPS of a mid-apron cyanobacterial mat, Orakei Korako, New Zealand. Scale bar, 2 mm. (**f**) Spherical to subspherical fossil bubbles (arrows) preserved in wavy laminated sinter, representing a silicified microstromatolite + EPS typical of mid-temperature sinter apron pools from the recently extinct Beowawe hot spring, Nevada. Scale bar, 200 μm. (**g**) Microbial palisade fabric with spherical structures, representing silicified gas bubbles in 15 Ka sinter, Tahunaatara, Taupo Volcanic Zone, New Zealand. Scale bar, 250 μm.

Therefore, regardless of whether the Dresser bubbles were derived from degassing of thermal fluids or represent metabolic gas derivatives, preservation likely occurred almost immediately, through entrapment in microbial EPS. Bubbles are commonly preserved in microbial sinter within mid-apron hot spring facies via trapping of microbial exudate (for example, oxygen)[15,16]. In modern examples, bubbles become silicified along with the microbial mat and either become infilled with sinter/microbial filaments or remain open[16]. Those forming in channels become flattened and appear almond shaped in cross-section[17], whereas bubbles formed in quiet, mid-temperature ($\sim$45–55 °C) apron pools may preserve spherical shapes[16] (Fig. 5f,g). The stratigraphic association of geyserite and horizons with bubbles in EPS can be explained as a function of Walther's Law, owing to laterally shifting discharge conditions or intermittent decreases in spring outflow temperatures, as vent geyserite can be found interbedded with mid- and low-temperature sinter apron fabrics[34].

In addition, within the unit of sinter terracettes, some thin laminae display vertically aligned quartz crystals (230 μm high) that wrap around the curved hinge of the convex ridges (Fig. 2c,d). Epithermal vein textures are discounted as there is no evidence of cross-cutting veins. Similarly, the vertically aligned fabric extends for many centimetres along bedding, whereas veins typically display quartz infill of an open cavity (Supplementary Fig. 9a). Shearing textures are also discounted since these would display quartz crystals aligned all in the same direction, rather than fanning around convex boundaries as is observed in the Dresser fabric (Supplementary Fig. 9b,c). Association with hot spring geyserite and the observation that the inferred palisade fabric is situated within interpreted sinter terracettes provide contextual support for formation in a hot spring setting. Therefore, these vertically aligned quartz crystals are suggested as analogous to recrystallized microbial palisade fabric formed through silicification of microbial filaments oriented perpendicular to bedding surfaces on mid- to distal-apron hot spring terraces[19] (Fig. 2e and Supplementary Fig. 9d).

Finally, a link between hot spring deposits and macroscopic stromatolites is drawn from geyserite rip-up clasts found interbedded with elongate domical and conical stromatolitic laminates (locality 24S: Supplementary Fig. 1) composed of ferruginized laminae (altered pyrite based on drill core comparisons[6]), and draped by ribbons and shards of felsic volcanic ash mixed with sand grains. Onlapping by these tuffaceous sediments and the irregular internal laminae with faint palimpsest microfabrics within these stromatolites are consistent with a microbial origin (Supplementary Fig. 10).

## Discussion

Textural similarity of the black-and-white laminated, laminar to botryoidal siliceous Dresser deposits with modern and Phanerozoic geyserite, combined with mineralogy consistent with geothermal settings, provide support for the previously unrecognized presence of geyserite in the Dresser Formation. Geyserite is known to precipitate from hot ($>$75–100 °C), silica-rich, near-neutral pH, alkali-chloride fluids ejected from boiling pools and geysers on exposed land surfaces[18,19]. A link between these surface fluids and the steam-heated acid sulfate alteration at the Dresser Mine[4] is provided by comparative mineralogy and studies of active geothermal systems. Steam-heated acid sulfate alteration forming kaolinite + illite mineral assemblages occurs in the late stages of an evolving high-sulfidation system[24]. In such a system, initial alkali-chloride geothermal springs subsequently develop into an acid steam-dominated system, commonly due to a drop in the water table under fluctuating, or waning, thermal activity[24]. In addition to geyserite, the discovery of siliceous sinter with terracettes and the mineralized remnants of hot spring pools collectively indicate a period of exposed land surface with terrestrial hot springs during Dresser deposition.

A hot spring setting is supported by the abrupt lateral and temporal facies changes observed over short distances (to the millimetre scale), including the patchy spatial distribution of geyserite, sinter and fluvial deposits. Inferred fluvial features include shallow channelized pebble to cobble conglomerate, distinctive edgewise conglomerate and cross-rippled sedimentary rocks. Such rapid changes are consistent with the 'tremendous variability observed in all siliceous hot springs'[34]. Variability is controlled by fluctuations in hot spring discharge, which influence temperature, flow rate, sinter facies and microbial growth[34]. This lateral and temporal variability contrasts markedly with Archaean marine or lacustrine deposits that are characterized by zoned stromatolite morphologies that typically show some degree of lateral and/or temporal continuity over tens,

to hundreds, and even thousands of metres[13,35]. Additionally, neither marine nor lacustrine settings can account for the preservation of geyserite, sinter terracettes or the mineralized remnants of hot spring pools.

Importantly, all recognized Dresser hot spring facies contain, or are spatially associated with, a suite of newly identified inferred biosignatures, including iron-rich domical and conical stromatolites, microbial palisade fabric within sinter terracettes and silicified bubbles in microbial EPS on sinter apron deposits. These observations suggest that early life in the Dresser Formation thrived off the chemical energy in hot springs.

In conclusion, newly discovered terrestrial hot spring facies in the ca. 3.5 Ga Dresser Formation contain a range of highly distinctive and varied textural biosignatures, providing direct evidence that at least some of Earth's earliest life thrived on land, in hot springs (Fig. 4). The Dresser Formation terrestrial hot spring facies include geyserite, siliceous sinter terracettes and the mineralized remnants of hot spring pools. These findings extend the geological record of inhabited terrestrial hot springs by ∼3 billion years, the occurrence of an exposed land surface by up to ∼130 million years[36,37] and evidence of life on land by ∼580 million years[21].

This result is significant in that it further constrains our understanding of the evolution of early life on Earth, as well as offers astrobiological implications in the search for potential fossil life on Mars. The Dresser Formation shares a similar age to older portions of the Martian crust and provides the closest comparison to geological processes likely occurring on Mars at that time[38]. The similarity of the Dresser deposits to modern hot springs shows that ancient hot spring processes on Earth were not so different from today. This lends weight to the use of modern and Phanerozoic terrestrial analogues in exploring for life in fossil Martian hot springs[19,39], and demonstrates the exceptional preservation potential of these very ancient fossil-bearing hot spring deposits here on Earth.

## Methods

**Mapping.** To assess possible associations between stromatolitic surface features and subsurface circulating hydrothermal fluids preserved in extensive vein networks, detailed mapping was undertaken to constrain geological context in a 7 km strike-length study area centred on the Dresser barite mine; 47 stratigraphic sections were measured. Voluminous amounts of barite were deposited as a result of extensive hydrothermal fluid circulation and therefore sedimentary units relatively proximal to these fluid conduits were chosen for in-depth investigation. Aerial photographs and geological maps of the area were used to navigate the terrain. GPS coordinates were collected from each locality for relocation.

**Light microscopy.** Optical light microscopy was conducted at the University of New South Wales on a Leica DM2500 with LAS 3.6 software to acquire photomicrographs.

**Raman spectroscopy.** Raman spectroscopy was measured at the Mark Wainwright Analytical Centre, UNSW using a Reni Shaw inVia Raman spectrometer using an argon ion laser providing excitation at 514 nm and fitted with an automated XYZ microscope stage. A 1 × 1 mm region of laminations on the petrographic thin section was selected using a ×50 objective and a series of measurements from 115–2,000 cm$^{-1}$ were collected at intervals of 25 μm. Laser spot sizes of ∼1 μm were estimated using this objective. Images were prepared using the Renishaw WiRE software by displaying the height of one major peak for each of the mineral phases of interest. Only two major mineral phases, quartz and anatase, were observed in the measured region. The spectra of quartz and anatase were examined, and one peak was selected from each spectrum that was unique to that mineral, with no overlap from other peaks. The intensity of quartz (red) was displayed using the peak at 484.8 cm$^{-1}$. The intensity of anatase (green) was displayed using the peak at 637.4 cm$^{-1}$. Images were blended, then superimposed on an incident white light image of the sample taken using a Leica M165C microscope.

**SEM-EDS.** Standard 30 μm thick petrographic thin sections were prepared with an evaporative carbon coating at the electron microscopy unit of the Mark Wainwright Analytical Centre at the University of New South Wales Australia. Samples were examined using a Hitachi S-3400N SEM operating at 20 kV and fitted with a Bruker SDD-EDS XFlash 6–30 detector.

**XRD.** *In situ* XRD analysis of the putative botryoidal geyserite was carried out by a Bruker D8 TXS with a basic parameter operating at 45 kV and 100 mA. The scan included 5–85 degrees (2Q). Micro-diffraction optics were used with a beam size of 0.5 mm.

**Data availability.** The authors declare that all data supporting the findings of this study are available within the paper (and its Supplementary Information files).

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

## Acknowledgements

Many thanks to: J. Reinter for discussion and assistance with Raman spectroscopic data; C. Marjo for assistance with Raman spectroscopic data; K. Privat for assistance with SEM-EDS data and the electron microscope unit, UNSW. Research support provided by the Australian Centre for Astrobiology and School of Biological, Earth and Environmental Sciences at the University of New South Wales, the Sloan Foundation and the ARC Centre for excellence Core to Crust Fluid Systems. Phanerozoic hot spring comparative studies were supported by funding to K.A.C. from the New Zealand government (RSNZ Marsden Fund and Ministry of Business, Innovation and Employment) and the National Geographic Society. Gigapan image generated by Ken Williford and the abcLab, Jet Propulsion Laboratory, California Institute of Technology. Kind hospitality in the field was provided by Faye and Geoff Myers, and Haoma Mining.

## Author contributions

The methodology was conceived and designed by T.D. and M.J.V.K. Geological mapping was carried out by T.D. and M.J.V.K. Petrographic analyses were carried out by T.D., M.J.V.K., K.A.C. and M.R.W. SEM-EDS data were acquired and interpreted by T.D. and M.J.V.K. XRD analysis spectra were acquired by C.R.W. All authors contributed to discussion, interpretation and writing.

## Additional information

**Competing interests:** The authors declare no competing financial interests.

DOI: 10.1038/ncomms16149 OPEN

# Corrigendum: Earliest signs of life on land preserved in ca. 3.5 Ga hot spring deposits

Tara Djokic, Martin J. Van Kranendonk, Kathleen A. Campbell, Malcolm R. Walter & Colin R. Ward

*Nature Communications* 8:15263 doi: 10.1038/ncomms15263 (2017); Published 9 May 2017; Updated 16 Aug 2017.

The original manuscript presented evidence that the deposits studied were fluvial rather than marine to support the conclusion that evidence of life on land had been discovered in the 3.4 Ga Dresser Formation, Pilbara Craton, Australia.

The authors acknowledge that the following reference, which was omitted, should have been cited to provide more detailed sedimentological and stratigraphic evidence that the deposits studied were indeed fluvial rather than marine.

Djokic, T. *Assessing the Link Between Earth's Earliest Convincing Evidence of Life and Hydrothermal Fluids: The c. 3.5 Ga Dresser Formation of the North Pole Dome, Pilbara Craton, Western Australia* (MPhil thesis, The University of New South Wales, 2015).

