## [Peer Review File · Nature Communications]

Editorial Note: In their review of the first version of this manuscript, reviewer 3 added their comments to a previous version of the manuscript. These comments, excluding minor textual revisions, have been copied into this Peer Review File.

Reviewers' Comments:

Reviewer #1 (Remarks to the Author)

The manuscript "Some liked it hot: earliest signs of life in ca. 3.5 Ga terrestrial hot spring deposits" proposes to have discovered biological features that supports the contention that life originated on land in association with hot springs. Although the authors provide convincing evidence that their sedimentary horizon (DFc1) is reminiscent of a hot spring deposit, my major concern about this work is the novelty, and even more so, the way that the authors try and sell their work. It is one thing to have discovered features of a hot spring at 3.48 Ga, it is quite another to try and convey that you have evidence pertaining to the origin of life. The fact that there are possible biological features in an ancient hot spring setting is interesting, but really how much of a scientific advance is it given the fact that a number of previous studies have already argued that the rocks represent an emergent hydrothermal system (i.e., the caldera model) and then used the same rocks to argue for life at that time, including the existence of microfossils and biologically fractionated stable isotopes. Further, the microfossil interpretation is rather weak, based on gas bubbles and sedimentary laminae. But, what is seriously flawed is that the authors try and link these features to the origin of life, something that is now suggested to have been at least 700 million years earlier. Given that the strength of this manuscript is the description of the hot spring features, not the biological features, my feeling is that this work would be better suited to a sedimentological journal.

Specific comments

7-10, The authors state that the Dresser has been reinterpreted as a volcanic caldera with "voluminous hydrothermal circulation" but then state that what is missing is evidence of surface manifestations of those hydrothermal fluids. What then was the evidence for the Dresser being subaerial? Also, if it was considered subaerial, why are we surprised that there are hot springs associated with it?

14-15, suggesting that this work is relevant to Mars missions is a questionable attempt of trying to sell the importance of this work given that much of the support for the Mars2020 mission is already based on microfossils preserved in modern hot spring cherts. Are the authors suggesting that putative fossil evidence from an ancient rock is more informative than studying the silicification processes in the modern?

19-22, what does origin of life have to do with this work? Reference 2 is a relatively old reference to use to build an argument against life having arisen in a hot environment, especially since there are a number of recent papers that argue for a hot origin.

28-30, if the caldera model already exists in the absence of hot spring evidence, then how important to the model is finding such deposits now?

36-38, biosignatures in rocks at 3.48 Ga do not provide support for an origin of life on land. Such a conflation of topics really undermines the possible significance of this work.

79-208, these sections convincingly demonstrate that DFc1 is likely to be a terrestrial hot deposit with preserved geyserite and subsurface barite deposits.

211-213, the authors state that they "augment previously reported circumstantial evidence suggesting life inhabited a volcanic-hydrothermal setting." This is disingenuous because the authors downplay previous work to then sell their own as being the more conclusive. This manuscript is certainly not the first to provide evidence for microbes in the Dresser – see the

authors own reference list.

222-265, As for their own "new evidence" it only consists of (i) potential gas bubbles that might suggest entrapment in EPS and (ii) laminae with convex structure and vertically aligned quartz crystals that are similar to microbial palisade fabrics. Therefore, the reader is left a bit confused as to what the new evidence is for microbial activity at 3.48 Ga, and how this possibly might expand our understanding of the origins of life. In fact, as the authors point out, geyserite by definition is a hot spring deposit that forms at temperatures between 100-750C, a temperature too hot to support microbial life other than hyperthermophiles. But in the intro the authors make a point of suggesting that life originated as a mesophile (line 21), so there appears to be a major disconnect between what the authors are arguing for.

Reviewer #2 (Remarks to the Author)

The results present new evidence for the antiquity of life and they are original since they examine spheres preserved within presumptive EPS that are likely to represent early biological material. One criticism I have is that the nature of those biosignatures could be stated in the abstract as that is quite important.

The data and methodology are appropriate. I think the authors should not mention specific missions as there is no guarantee that they will succeed.

The conclusions are robust and highly novel.

One major criticism I have is the mention of the origin of life. The fact that there was life in this terrestrial setting does not provide evidence that it was a setting for the origin of life. The assembly of the first prebiotic compounds into a self-replicating cell may have required entirely different conditions. The idea that the findings support an early land origin of life I think distracts from the main point of the work.

line 84. The clarity of 'presence of rip-off clasts in edgewise conglomerates' could be improved as this is quite important for the sedimentary interpretation.

lines 133 and 138. In 133, it says temperatures less than 100C in 138 it says not more than 400C yet that leaves about 300C where the temperature could have been, but in which geysers have not been found. Could the temperature still have been too high for the geysers that we know?

line 203. Some more detail on how the barites compare to those of Lake Bogoria, Kenya would be useful since this seems critical for the interpretation.

Reviewer #3 (Remarks to the Author)

Comments on attached marked up manuscript

Reviewers' comments:

Note: Highlighted in green are minor points not addressed in the cover letter.

Reviewer #1 (Remarks to the Author):

The manuscript "Some liked it hot: earliest signs of life in ca. 3.5 Ga terrestrial hot spring deposits" proposes to have discovered biological features that supports the contention that life originated on land in association with hot springs. Although the authors provide convincing evidence that their sedimentary horizon (DFc1) is reminiscent of a hot spring deposit, my major concern about this work is the novelty, and even more so, the way that the authors try and sell their work. It is one thing to have discovered features of a hot spring at 3.48 Ga, it is quite another to try and convey that you have evidence pertaining to the origin of life.

We agree that the Dresser Formation is not the site of the origin of life and have no intention of proposing this. It was to highlight the fact that on Early Earth there existed an environmental context consistent with one of the current origin of life hypotheses being debated, that of terrestrial hot springs. What is apparent from the review process is that reflection on origin of life studies in this manuscript detracts from the significance of this work. We do not want to encounter any further misunderstanding, so have removed all reference to the origin of life from the manuscript. We also have modified the title of the manuscript to focus on the key finding of our work.

The fact that there are possible biological features in an ancient hot spring setting is interesting, but really how much of a scientific advance is it given the fact that a number of previous studies have already argued that the rocks represent an emergent hydrothermal system (i.e., the caldera model) and then used the same rocks to argue for life at that time, including the existence of microfossils and biologically fractionated stable isotopes.

No one has shown direct evidence that it was emergent.

No one has shown that the life was directly associated with hot springs.

No one has given direct evidence of life living at surface hot springs so long ago – only indirect evidence from fluid inclusions has suggested hot springs, but no life was directly correlated with this.

Further, the microfossil interpretation is rather weak, based on gas bubbles and sedimentary laminae.

.....

But, what is seriously flawed is that the authors try and link these features to the origin of life, something that is now suggested to have been at least 700 million years earlier. Given

that the strength of this manuscript is the description of the hot spring features, not the biological features, my feeling is that this work would be better suited to a sedimentological journal.

We agree that the Dresser Formation is not the site of the origin of life. To avoid any further misunderstanding and as not to detract from the significance of this work, all reference to the origin of life has been removed. Furthermore, we have addressed the bubble interpretation with text clarifications and a new comparative modern photograph.

Reviewer #1 specific comments:

7-10, The authors state that the Dresser has been reinterpreted as a volcanic caldera with “voluminous hydrothermal circulation” but then state that what is missing is evidence of surface manifestations of those hydrothermal fluids. What then was the evidence for the Dresser being subaerial? Also, if it was considered subaerial, why are we surprised that there are hot springs associated with it?

Previous interpretations of the Dresser Fm used sedimentological observations including dessication cracks as well as low temperatures for steam heated acid sulphate altered pillow basalts, also indicated by fluid inclusion data to suggest a subaerial caldera setting with hot springs. However, given that conflicting palaeoenvironmental interpretations still exist i.e. recently suggested as a tidal flat (Noffke *et al.* 2013), the observation of hot spring deposits directly precipitating at the surface alongside biosignatures unequivocally supports an emergent caldera as opposed to a quiet shallow marine setting. The key here is clear evidence that life lived on land in the early Archean in hot springs, as opposed to a marine-based setting or within subsurface hydrothermal vents – which is what has previously been reported. We should not be surprised that hot spring deposits existed in this setting, however the point here is that prior to this work they have not been documented i.e. no direct evidence for life living in and around the surface at hot spring pools.

Noffke, N., Christian, D., Wacey, D. & Hazen, R. M. Microbially induced sedimentary structures recording an ancient ecosystem in the ca. 3.48 billion-year-old Dresser Formation, Pilbara, Western Australia. *Astrobiology* **13**, 1103-1124 (2013).

Line 14-15: suggesting that this work is relevant to Mars missions is a questionable attempt of trying to sell the importance of this work given that much of the support for the Mars2020 mission is already based on microfossils preserved in modern hot spring cherts. Are the authors suggesting that putative fossil evidence from an ancient rock is more informative than studying the silicification processes in the modern?

On the contrary, the point is that the Dresser is very close in age to most of the Martian crust, and as such, gives the closest comparison to processes at that time. Previously the oldest known hot spring deposits were in the Devonian, a time very distant from the time on Mars when volcanoes were active and water was flowing on the surface. On Earth, volcanoes + water produces hot springs teeming with microbial life. Hence, hydrothermal settings are important candidate sites for future Mars lander-rover missions. The similarity of Dresser deposits to modern hot springs shows that ancient processes were not so different from today, so it actually lends weight to using Archean and modern analogues to explore for life on Mars. Thus, this discovery will be of major importance to the Mars community. This has been clarified in the text.

Line 19-22: what does origin of life have to do with this work? Reference 2 is a relatively old reference to use to build an argument against life having arisen in a hot environment, especially since there are a number of recent papers that argue for a hot origin.

It was to highlight the fact that on Early Earth there existed an environmental context consistent with one of the current origin of life hypotheses being debated, that of terrestrial hot springs. What is apparent is that reflection on origin of life studies in this manuscript detracts from the significance of this work. We do not want to encounter any further misunderstanding, so have removed all reference to the origin of life from the manuscript.

Line 28-30: if the caldera model already exists in the absence of hot spring evidence, then how important to the model is finding such deposits now?

The caldera model itself does not provide actual examples of the types of textures – physical and biological – that would be expected to be preserved at the surface in and around the hot springs themselves. This work provides these examples in the form of a range of biosignatures, several of which have not previously been reported from the Dresser Formation.

Line 36-38: biosignatures in rocks at 3.48 Ga do not provide support for an origin of life on land. Such a conflation of topics really undermines the possible significance of this work.

There was certainly no intention to conflate topics of origins of life. Nonetheless, considering that burgeoning research in origin of life studies is seriously arguing for terrestrial hot springs as possible sites for the origins of life (Deamer & Georgiou, 2015) and that opposing theories (deep sea vents) use the argument that little or no land was available

in the early Archean (Barge *et al.* 2016), the geological and geochemical context provided by the Dresser Formation may offer some insight into relevant environmental conditions. However, it is agreed that the significance of this work is not focussed on the origins of life. Therefore all references to the origin of life have been removed in order to avoid any misunderstanding.

Barge, L. *et al.* Thermodynamics, Disequilibrium, Evolution: Far-From-Equilibrium Geological and Chemical Considerations for Origin-Of-Life Research. *Origins of Life and Evolution of Biospheres*, 1-18 (2016).

Deamer, D. W. & Georgiou, C. D. Hydrothermal Conditions and the Origin of Cellular Life. *Astrobiology* **15**, 1091-1095 (2015).

Line 79-208: these sections convincingly demonstrate that DFC1 is likely to be a terrestrial hot spring deposit with preserved geyselite and subsurface barite deposits.

Great.

Line 211-213: the authors state that they “augment previously reported circumstantial evidence suggesting life inhabited a volcanic-hydrothermal setting.” This is disingenuous because the authors downplay previous work to then sell their own as being the more conclusive. This manuscript is certainly not the first to provide evidence for microbes in the Dresser – see the authors own reference list.

There are of course many convincing previous reports of life in Dresser, but none provide a direct correlation between life and terrestrial hot spring deposits. We have reworded this section for clarity as we did not intend to downplay previous work, but rather aim to show how this current research supports and extends earlier findings.

Line 222-265: ‘ own “new evidence” it only consists of (i) potential gas bubbles that might suggest entrapment in EPS and (ii) laminae with convex structure and vertically aligned quartz crystals that are similar to microbial palisade fabrics. Therefore, the reader is left a bit confused as to what the new evidence is for microbial activity at 3.48 Ga, and how this possibly might expand our understanding of the origins of life.

It’s not that this is the best evidence of life in the Dresser, or the only evidence, but the earliest evidence for life in hot spring deposits on land, which has not previously been reported. This is the first time there has been documentation of the occurrence of Archean biosignatures directly related to terrestrial hot springs. This has been clarified in the text. In addition it is apparent that reflections on origins of life detracts from the significance of this

work. Therefore, all references to the origins of life have been removed.

In fact, as the authors point out, geysers by definition are hot spring deposits that form at temperatures between 100-75°C, a temperature too hot to support microbial life other than hyperthermophiles. But in the intro the authors make a point of suggesting that life originated as a mesophile (line 21), so there appears to be a major disconnect between what the authors are arguing for.

We are not suggesting the Dresser Formation is the site of the origin of life. The observation of geysers provides direct evidence for terrestrial hot springs. We have removed all reference to origins of life to avoid any further misunderstanding.

Reviewer #2 (Remarks to the Author):

The results present new evidence for the antiquity of life and they are original since they examine spheres preserved within presumptive EPS that are likely to represent early biological material. One criticism I have is that the nature of those biosignatures could be stated in the abstract as that is quite important.

The data and methodology are appropriate. I think the authors should not mention specific missions as there is no guarantee that they will succeed.

This has been revised in the text

The conclusions are robust and highly novel.

One major criticism I have is the mention of the origin of life. The fact that there was life in this terrestrial setting does not provide evidence that it was a setting for the origin of life. The assembly of the first prebiotic compounds into a self-replicating cell may have required entirely different conditions. The idea that the findings support an early land origin of life I think distracts from the main point of the work.

There was certainly no intention to suggest the Dresser Formation as the site of the origin of life. Considering that burgeoning research in origin of life studies is seriously arguing for terrestrial hot springs as possible sites for the origins of life (Deamer & Georgiou, 2015) and that opposing theories (deep sea vents) use the argument that little or no land was available in the early Archean (Barge *et al.* 2016), the geological and geochemical context provided by the Dresser Formation may offer some insight into relevant environmental conditions. However, it is agreed that the significance of this work is not focussed on the origins of life.

Therefore all references to the origin of life have been removed in order to avoid any misunderstanding.

Barge, L. *et al.* Thermodynamics, Disequilibrium, Evolution: Far-From-Equilibrium Geological and Chemical Considerations for Origin-Of-Life Research. *Origins of Life and Evolution of Biospheres*, 1-18 (2016).

Deamer, D. W. & Georgiou, C. D. Hydrothermal Conditions and the Origin of Cellular Life. *Astrobiology* **15**, 1091-1095 (2015).

Line 84: The clarity of 'presence of rip-off clasts in edgewise conglomerates' could be improved as this is quite important for the sedimentary interpretation.

This has been clarified in the text.

Lines 133 and 138: In 133, it says temperatures less than 100C in 138 it says not more than 400C yet that leaves about 300C where the temperature could have been, but in which geysers have not been found. Could the temperature still have been too high for the geysers that we know?

Good question. Temperature constraints from Harris *et al.* (2009) indicate cool 120 degree C waters for DFC1, supporting lower temperatures for the units containing geysers. This has now been cited in the text.

Harris, A. C. *et al.* Early Archean hot springs above epithermal veins, North Pole, Western Australia: new insights from fluid inclusion microanalysis. *Economic Geology* **104**, 793-814 (2009).

Line 203: Some more detail on how the barites compare to those of Lake Bogoria, Kenya would be useful since this seems critical for the interpretation.

Barite is not present in Lake Bogoria, but the textures/structures of cavity fill are similar, which is why this reference has been provided - to present one possible formation process. This has been clarified in the text.

Reviewer #3 (Remarks to the Author):

Line 18-24: The 3.5 Ga rocks should not be directly linked with the origin of life. Rocks in the Pilbara regions are likely much younger than the origin of life itself. This is not about the origin of life, it is about biosignatures in a subaerial hydrothermal environment, which is the most important discovery in this paper. To me the origin of life here is out of context...

Reference to origins of life has been removed.

Line 30: Previous studies have shown the presence of micritic carbonate units interbedded with volcanoclastic sediment below the bedded chert... Hydrothermal fluid circulation must have interacted with those units. Travertine would be expected to form at the surface... are there any hints pointing to travertine formation?

This would be important for the diversity of environments and processes that could be present at the time –

There is current research being undertaken on this, but is still under analyses and not yet ready for publication.

Line 33: This term implies necessarily a source of flow discharge. You could have microlaminated sinter without having a discharge, in which case it would be simply a sinter. For modern geysers, the vent needs to be present. I can see that the sinter can be evidenced in the Dresser Fm, but I cannot see anything that would point specifically to a vent/chimney/geyser.

At the barite curl we have provided evidence for the edge of a hot spring pool (i.e. collapsed pool wall), which provides evidence for a vent. At the geysere locality, barite veins end at the level of the geysere and are thus inferred as the mineralized remnants of a vent. These points have been clarified in the text.

Also see below in response to the comment regarding microlaminated sinter.

Line 72-73: however, the consensus is that you need a discharge/vent in order to call something a geysere... otherwise it would be simply a sinter deposit. Microlamination is not exclusive of geyseres but common to most sinters. I suggest caution when using those terms, to prevent misleading those working specifically on geysers and geysere structures.

All sinters away from vents do not have extremely fine, dense lamination (like geysere) as they are dominated by microbial mats, which cause much more irregular laminae within biofilms and stromatolites (Lynne, 2012). The really fine lamination is typical of true geysere forming at hot spring vent environments. In addition, we have provided multiple lines of evidence that support a geysere origin, not just microlamination.

Lynne, B. Y. Mapping vent to distal-apron hot spring paleo-flow pathways using siliceous sinter architecture. *Geothermics* **43**, 3-24 (2012).

Line 76: and also, microbes (their biomass) can influence the morphological development of the resulting sedimentary structures

This has been clarified in the text.

Line 77: unless rapid silicification of cellular materials occurs (see papers by Jack D Farmer and Sherry L Cady)... see also Konhauser et al, 2001, Sedimentology. for a discussion on the potential preservation of microbial structures in sinters.

I will not deny K Campbell's statements on this regard, but it cannot be generalized that organic materials and structures will always be lost through diagenetic processes

This has been clarified in the text.

Line 94: this also could happen down the flow and away from any vent/discharge. Thus, sintraclasts and infilled troughs are not diagnostic for geysers solely.

It appears the reviewer might have misinterpreted the observations. The infilled troughs are micro-features within the columnar to botryoidal geysers, not macro sedimentary features. While the sintraclasts are not intended to argue for a geysers origin, but for the syn-depositional nature of the deposits – This has been clarified in the text.

Alternately, we may have we may have misunderstood the reviewer's concern – nevertheless, we have tried to clarify the issue.

Line 106: but also sinters away from the discharge source

The reviewer's comment is in regards to microlaminated features in the Dresser geysers – suggesting that these features are also found in sinters away from the discharge source. As stated above - All sinters away from vents do not have extremely fine lamination as they are dominated by microbial mats, which cause much more irregular laminae within biofilms and stromatolites. The really fine lamination is typical of true geysers forming at hot spring vent environments. In addition, we have provided multiple lines of evidence that support a geysers origin, not just microlamination.

Line 119: suppl figs 1-5

Error in figure numbers updated to 2-6 in text.

Line 156: this is precisely my point. All the evidence is suggestive of geysers deposits, but no discharge structures are found. Therefore, in my opinion, these deposits are 100 C to ambiente temperature sinters, where high-energy discharges (geysers) were likely present in the surroundings, but the sinter deposits are not strictly indicative of geysers.

The geysers rip-up clasts are included to make a point about the syn-depositional nature of the geysers deposits. This does not detract from the fact that it is geysers that formed at a vent, which was later reworked and redeposited.

It is important to emphasize some of the Dresser hot spring settings were fluvial. It is also important to note that hot spring vent areas do not have to be huge, as in many modern hot spring examples, and in the case of the Dresser no larger than the barite curl/vent pool

dimensions and with some smaller and quite localized occurrences, e.g. the geyserite locality.

Not all vent related geyserite forms under vigorous surging and splashing conditions. Deena Braunstein and Don Lowe showed this at Yellowstone.

Braunstein, D. & Lowe, D. R. Relationship between spring and geyser activity and the deposition and morphology of high temperature (> 73 C) siliceous sinter, Yellowstone National Park, Wyoming, USA. *Journal of Sedimentary Research* **71**, 747-763 (2001).

As mentioned above: At the barite curl we have provided evidence for the edge of a hot spring pool (i.e. collapsed pool wall), which provides evidence for a vent. At the geyserite locality, barite veins end at the level of the geyserite and are thus inferred as the mineralized remnants of a vent.

These points have been clarified in the text.

Line 161: This is not a strict rule. I have explored geothermal fields with active sinter deposition and the water table is >6 m below the top ground surface.

This has been addressed in the text by replacing the word 'only' by 'often'

Line 192: are you describing a sinkhole? empty pools and collapsed sinter may form them...

No. It is above hydrothermal veins and is not symmetrical.

Sink holes are generally round

Line 207: This reference does not report subaerial barite, but shallow marine... but you can cite Bonny & Jones, 2008, *Sedimentology*...

Regards. This has been amended.

Line 212: stick to one term all throughout the text... either terrestrial or subaerial

All subaerial hot springs have been updated in the text to terrestrial for consistency.

Line 214: what about sinters and stromatolites?

We have added silica sinter and stromatolites in the text for clarification.

Line 231: They also look like acritarchs... read Wilmeth et al, 2015, *Palaios* where they describe the evolution of gas bubbles within Cambrian oncolites. Their bubbles look very different from yours.

Regarding acritarchs: What we interpret as bubbles have none of the characteristic features of acritarchs such as spines or the presence of kerogen. Considering this, we do not see it essential to argue against this in the text.

Regarding Wilmeth et al. 2015: Yes the bubbles in the oncolites are deformed by a carbonate mud/microbe medium. In our case, and in the case of the bubbles we observed *in situ* at Orakei Korako (Figure 4e), we only see the bubbles themselves, generated as a microbial metabolic by-product, presumably *in situ*, with little or no movement, perhaps due to the extra sticky nature of the EPS matrix in the hot spring microbial mats. This has been described in the text and the paper in question added as a reference. An additional figure has also been added to show examples of bubbles preserved in more recent sinter.

Line 258-260: I agree with your interpretation, but other mechanisms are possible, for example oriented growth of quartz outward from the surface, such as colloform or comb-like textures. See Dong et al, 1995, Economic Geology –

The context is consistent with these being terracettes and in outcrop there is no evidence of cross cutting veins. This has been clarified in the in the text

Line 274-277: why would you expect an extensive outcrop? most geothermal fields today are localized and short ranged - relatively –

Indeed that is correct. We don't expect/suggest an extensive outcrop. Abrupt lateral facies changes is exactly what we have observed and described in the text. We have clarified the text to address any misunderstanding.

Line 290-293: I do not agree. This paper emphasizes that hydrothermal systems with active sinter-forming processes existed in the past. This helps arguing that other hydrothermal systems (geysers, travertines and all the different depositional settings that we see today) likely existed as well... this enlarges our view of the scenarios where life thrived in the past... Life may have originated (also, but not exclusively) in these type of environments, that is fine, but the origin of life involves other chemical and physical processes that are not directly related to this work. –

It is true that we don't know the chemical and physical processes of the origins of life. We are not saying that this is a direct representative. What we are suggesting is that these habitats were indeed available on early Earth. This supports one potential environmental model for life's origins – but indeed not the origin point itself. All reference to origins of life has been removed to avoid any further misunderstanding. We appreciate that this should not detract from the significance of this paper.

Figure 4: where is the original image of this spectrum? it is difficult to interpret the spatial arrangement of the 'bubble' from this image...

Image updated to show orientation of the bubble with respect to anatase crystal splay.

Figure 5a: please make sure that it is clearly said that these are not ripple marks. The photo is not very illustrative in that regard

nature communications_Full list of response to reviewer comments

This is clearly stated in the text.

Reviewers' comments:

Reviewer #1 (Remarks to the Author):

This is the second time that I have reviewed this manuscript. I appreciate that in this revisions the authors have removed their discussions of the origins of life and have clarified a number of the points I raised earlier. However, there are still two issues that prevent me from signing off on this manuscript, and I leave it to the Editor to decide if my concerns should, or should not, preclude publication.

First, I initially questioned the significance of these findings, and after reviewing this version, I still have the same concerns. What I initially wrote, and to which the authors were dismissive, is "the fact that there are possible biological features in an ancient hot spring setting is interesting, but really how much of a scientific advance is it given the fact that a number of previous studies have already argued that the rocks represent an emergent hydrothermal system (i.e., the caldera model) and then used the same rocks to argue for life at that time, including the existence of microfossils and biologically fractionated stable isotopes." The authors reply: "No one has shown direct evidence that it was emergent. No one has shown that the life was directly associated with hot springs. No one has given direct evidence of life living at surface hot springs so long ago – only indirect evidence from fluid inclusions has suggested hot springs, but no life was directly correlated with this".

With regards to point 1, Buick et al. (1995) in their paper "Record of emergent continental crust ~3.5 billion years ago in the Pilbara craton of Australia" discuss the subaerial nature of rocks closely associated with those here.

With regards to points 2 and 3, Harris et al. (2009) stated the following "Our findings reveal that the earliest life known on Earth lived in and around a hydrothermal system with temperatures from ~300°C at depth to 120°C near the paleosurface, in an environment closely analogous to modern hot springs". So what if the evidence is from fluid inclusions? The point is that these rocks being described as hot spring deposits is not new.

Also, the argument of very shallow marine versus hot spring is rather semantic because Van Kranendonk et al. (2008) suggested that "the silica ± barite veins that underlie the bedded chert-carbonate units are interpreted as fossil hydrothermal fluid pathways that formed synchronously with, and were the cause of, deposition of the bedded chert-barite units. Hydrothermal fluid flow was accompanied by extensive hydrothermal alteration of the footwall, with characteristic assemblages and zonation indicative of a steam-heated acid-sulfate hydrothermal system under shallow water conditions". Further citing Van Kranendonk et al. (2008); "..... intense hydrothermal alteration (including acid-sulfate alteration) that has affected footwall rocks beneath the bedded chert-barite units, but not the overlying rocks (Ueno et al., 2001b; Van Kranendonk and Pirajno, 2004; Pirajno and Van Kranendonk, 2005; Van Kranendonk, 2006). These authors provided clear evidence that hydrothermal activity occurred during accumulation of the sedimentary pile in the form of syn-depositional hydrothermal breccias and footwall alteration." So, the difference comes down to very shallow marine versus emergent. Plus, the title of that paper is "Geological setting of Earth's oldest fossils in the ca. 3.5 Ga Dresser Formation, Pilbara Craton, Western Australia", so the argument that the life aspect is new is misleading.

Second, I stated that the microfossil interpretation is rather weak, and it only consists of (i) potential gas bubbles that might suggest entrapment in EPS and (ii) laminae with convex structure and vertically aligned quartz crystals that are similar to microbial palisade fabrics. To this the authors reply with "It's not that this is the best evidence of life in the Dresser, or the only evidence, but the earliest evidence for life in hot spring deposits on land, which has not previously been reported. This is the first time there has been documentation of the occurrence of Archean biosignatures

directly related to terrestrial hot springs.”

So, is it fair to conclude that maybe the evidence for life presented here is not definitive proof of ancient life in these rocks? If so, doesn't this undermine their main premise of showing that life existed in a hot spring setting at 3.5 Ga?

Reviewer #2 (Remarks to the Author):

No further comments

Reviewer #3 (Remarks to the Author):

I agree with the corrections made by the authors. I was one of the ones arguing against statements involving the origin of life, and I think the authors now agree with us, and we with them.

Response to reviewers' comments

Reviewer #1 (Remarks to the Author):

This is the second time that I have reviewed this manuscript. I appreciate that in this revisions the authors have removed their discussions of the origins of life and have clarified a number of the points I raised earlier. However, there are still two issues that prevent me from signing off on this manuscript, and I leave it to the Editor to decide if my concerns should, or should not, preclude publication.

First, I initially questioned the significance of these findings, and after reviewing this version, I still have the same concerns. What I initially wrote, and to which the authors were dismissive, is “the fact that there are possible biological features in an ancient hot spring setting is interesting, but really how much of a scientific advance is it given the fact that a number of previous studies have already argued that the rocks represent an emergent hydrothermal system (i.e., the caldera model) and then used the same rocks to argue for life at that time, including the existence of microfossils and biologically fractionated stable isotopes.” The authors reply: “No one has shown direct evidence that it was emergent. No one has shown that the life was directly associated with hot springs. No one has given direct evidence of life living at surface hot springs so long ago – only indirect evidence from fluid inclusions has suggested hot springs, but no life was directly correlated with this”.

With regards to point 1 (i.e. No one has shown direct evidence that it was emergent), Buick *et al.* (1995) in their paper “Record of emergent continental crust ~3.5 billion years ago in the Pilbara Craton of Australia” discuss the subaerial nature of rocks closely associated with those here.

The Buick *et al.* (1995) paper dealt with an entirely different, and considerably younger (i.e. up to ~130 million years younger) formation, the Strelley Pool Formation, deposited ca. 4.43 - 3.35 Ga. This unit was deposited in a very different setting from the Dresser Formation, as a craton-wide shallow marine deposit formed on an erosional unconformity (Van Kranendonk *et al.*, 2002; 2007).

With regards to points 2 and 3 (i.e. No one has shown that the life was directly associated with hot springs. No one has given direct evidence of life living at surface hot springs so long ago – only indirect evidence from fluid inclusions has suggested hot springs, but no life was directly correlated with this), Harris *et al.* (2009) stated the following “Our findings reveal that the earliest life known on Earth lived in and around a hydrothermal system with temperatures from ~300°C at depth to 120°C near the paleosurface, in an environment closely analogous to

modern hot springs”. So what if the evidence is from fluid inclusions?

We appreciate that the reviewer indicates that a model of subaerial hot springs had already been suggested, but – as we stated in our previous response – no previous authors, including Harris *et al.* (2009), provided any direct evidence that the subterranean hydrothermal veins debouched onto the surface, or that life thrived in, and around the peripheries of, surface hot springs, as we have done in this contribution. Rather, Harris *et al.* (2009) inferred that the setting was “closely analogous” to hot springs and “near the surface”, based on examination of fluid inclusions from subsurface veins, not surficial deposits, and so, the results were not diagnostic of an exposed land surface inhabited by life. Moreover, although the identification of a caldera setting described by previous authors (e.g., Van Kranendonk and Pirajno, 2004; Van Kranendonk, 2006; Van Kranendonk *et al.* 2008) document periods of shallow water and locally exposed or subaerial conditions, they do not report any deposits that represent an exposed land surface exclusively of a terrestrial, and not marine, setting. Submarine calderas are well documented in the literature (e.g., de Ronde *et al.*, 2005) in addition to those that form subaerially. In our manuscript, we have shown that life was categorically living in and around active hot springs on an exposed land surface by placing the microbial facies side-by-side with those of terrestrial hot spring deposits.

The significance to what may appear to this reviewer (but not the other two reviewers) as a pedantic difference between shallow marine and exposed conditions in these ancient rocks is actually vitally important to understanding the evolution of early life and therefore has astrobiological implications. Life’s adaptation to land (or at least to fresh water) is generally considered not to have occurred until the late Archean, at ca. 2.7 Ga (Beraldi-Campesi, 2013) and this is assumed as evidence for a late adaptation of life to terrestrial conditions. However, this work suggests the opposite is true, and so if life ever developed in a similar way on Mars this work provides the best representation of what we might find in ancient Martian deposits. We have endeavored to make this clearer in the MS.

The point is that these rocks being described as hot spring deposits is not new.

This is not true. As outlined above, no one has previously unambiguously provided evidence of deposition from hot springs in a terrestrial setting in the Archean, in fact earlier than Devonian – our documentation of geyserite does so, augmented by our recognition of siliceous sinter and our interpretation that the large, geometrically unusual barite masses are not seafloor domes, but the mineralised remnants of the hot spring pools themselves. These are completely new discoveries and absolutely verify a terrestrial setting. The further discovery

of new and diverse types of biosignatures preserved within these rocks is also completely novel and important, as it ties a surprisingly diverse suite of life forms firmly to a terrestrial setting.

The significance here is not just about support for a terrestrial hot spring model for the Dresser Formation - although this work has provided a smoking gun (i.e. geyserite) for this model that was previously undocumented - but that exceptional preservation has occurred in such ancient deposits. The 'so what factor' is that the Dresser Formation rocks are unarguably the most closely analogous to those on Mars (being the same age to much of the Martian crust) and preserve biosignatures. Critically, the microfacies identified in the Dresser Formation are similar to those from modern hot springs on Earth, and hot spring deposits with Earth analogies have been found on Mars (Ruff and Farmer, 2016). This provides strong support for the potential preservation of biosignatures on Mars in fossil hot springs systems. Thus, our study provides real world (geologically parallel time scales) examples of the potential biosignatures that might exist in ancient Martian hot springs.

Also, the argument of very shallow marine versus hot spring is rather semantic because Van Kranendonk et al. (2008) suggested that "the silica ± barite veins that underlie the bedded chert-carbonate units are interpreted as fossil hydrothermal fluid pathways that formed synchronously with, and were the cause of, deposition of the bedded chert–barite units. Hydrothermal fluid flow was accompanied by extensive hydrothermal alteration of the footwall, with characteristic assemblages and zonation indicative of a steam-heated acid–sulfate hydrothermal system under shallow water conditions". Further citing Van Kranendonk et al. (2008); "... intense hydrothermal alteration (including acid–sulfate alteration) that has affected footwall rocks beneath the bedded chert–barite units, but not the overlying rocks (Ueno et al., 2001b; Van Kranendonk and Pirajno, 2004; Pirajno and Van Kranendonk, 2005; Van Kranendonk, 2006). These authors provided clear evidence that hydrothermal activity occurred during accumulation of the sedimentary pile in the form of syn-depositional hydrothermal breccias and footwall alteration." So, the difference comes down to very shallow marine versus emergent. Plus, the title of that paper is "Geological setting of Earth's oldest fossils in the c. 3.5 Ga Dresser Formation, Pilbara Craton, Western Australia", so the argument that the life aspect is new is misleading.

As we outlined in our first response above, the difference between shallow marine conditions and an exposed land surface is significant for our understanding of the evolution of life on Earth, but also for the search for life on Mars. Similarly, although Van Kranendonk *et al.* (2008) documented a contemporaneous deposition of sediment and formation of hydrothermal veins, they do not discriminate between shallow marine and terrestrial conditions. In this manuscript, it is the evidence for emergence of land and the tie-in with preserved

biosignatures that is significant.

Second, I stated that the microfossil interpretation is rather weak, and it only consists of (i) potential gas bubbles that might suggest entrapment in EPS and (ii) laminae with convex structure and vertically aligned quartz crystals that are similar to microbial palisade fabrics. To this the authors reply with “It’s not that this is the best evidence of life in the Dresser, or the only evidence, but the earliest evidence for life in hot spring deposits on land, which has not previously been reported. This is the first time there has been documentation of the occurrence of Archean biosignatures directly related to terrestrial hot springs.”

Our reply was with respect to the concluding remarks (underlined below) in that specific comment. The full comment by the reviewer, in the previous review, read: **“Line 222-265:** own “new evidence” it only consists of (i) potential gas bubbles that might suggest entrapment in EPS and (ii) laminae with convex structure and vertically aligned quartz crystals that are similar to microbial palisade fabrics. Therefore, the reader is left a bit confused as to what the new evidence is for microbial activity at 3.48 Ga, and how this possibly might expand our understanding of the origins of life.”

So, is it fair to conclude that maybe the evidence for life presented here is not definitive proof of ancient life in these rocks? If so, doesn’t this undermine their main premise of showing that life existed in a hot spring setting at 3.5 Ga?

It would be unreasonable to suggest that definitive proof of life is categorically achievable when dealing with such ancient rocks. However, there are three aims of this manuscript: 1) provide new, diagnostic evidence of terrestrial hot spring deposits in the Dresser Formation; 2) that life was associated with these deposits; 3) argue that this work is part of a bigger picture effort to better understand the development of early life on Earth in pursuit of an improved measure in the search for life on Mars.

We have provided observations that to the best of our deductions have achieved these objectives, providing previously undocumented hot spring deposits, which are important for reasons stated above.

Specifically regarding the biosignatures, Reviewer 1 states that the microfossil interpretation is rather weak”. We present evidence of the preservation of microbial activity that two other reviewers have found to be compelling, rather than weak. Without any justification from this reviewer as to why that evidence is weak, we are unable to address his/her concerns. However, we endeavor to present the best illustration of the evidence, therefore we have added/revised:

- Supplementary Fig. 9 and revised Fig. 2e (formally Fig. 5e) to underscore the parallels between the Dresser Formation palisade textures and those produced by microbial activity in Recent hot spring deposits.
- Additional part, Fig. 5g (formally Fig. 4), which clearly shows preserved gas bubbles in more recent hot spring deposits, trapped by microbial EPS.

Moreover, our previous comments relating to life in the Dresser Formation were simply that many of the biosignatures we present in this manuscript are new and different to those previously documented (stromatolites, fractionated stable isotopes). In combination, they add to the growing body of evidence for not only life at this ancient terrestrial hydrothermal setting, but diverse life (e.g. Van Kranendonk, 2011). We suggest this is a startling new finding given the antiquity of the deposits, and their similarities to modern hot spring biosignatures.

Critically, the geological context we provide here provides immense support to our interpretation of preserved biosignatures, as they occur within microfacies emblematic of those in modern hot springs (i.e. association of botryoidal geyserrite, bubbles and channelized stromatolitic laminates (Supplementary Fig. 1c); association of palisade fabric with sinter terracettes and stratiform geyserrite (Supplementary Fig. 1d); association of stromatolites with geyserrite rip-up clasts (Supplementary Fig. 1e).

References absent from Manuscript

de Ronde, C. E. J., Hannington, M. D., Stoffers, P., Wright, I. C., Ditchburn, R. G., Reyes, A. G., Baker, E. T., Massoth, G. J., Lupton, J. E., Walker, S. L., Evolution of a submarine magmatic-hydrothermal system: Brothers volcano, southern Kermadec arc, New Zealand. *Economic Geology* **100**, 1097-1133 (2005)

Pirajno, F. & Van Kranendonk, M. Review of hydrothermal processes and systems on Earth and implications for Martian analogues. *Australian Journal of Earth Sciences* **52**, 329-351 (2005).

Ueno, Y., Maruyama, S., Isozaki, Y. & Yurimoto, H. Early Archean (ca. 3.5 Ga) microfossils and ¹³C-depleted carbonaceous matter in the North Pole area, Western Australia: Field occurrence and geochemistry. *Geochemistry and the Origin of Life*, 203-236 (2001b).

Van Kranendonk, M. J., Hickman, A. H., Smithies, R. H., Nelson, D. R. & Pike, G. Geology and tectonic evolution of the archaic North Pilbara terrain, Pilbara Craton, Western Australia. *Economic Geology* **97**, 695-732 (2002).